# Decreased risk of underdosing with continuous infusion versus intermittent administration of cefotaxime in patients with sickle cell disease and acute chest syndrome

**Keyvan Razazi**[1,2,3☉]*, **Enora Berti**[1,3☉], **Jerome Cecchini**[1,3,4], **Guillaume Carteaux**[1,2,3], **Anoosha Habibi**[2,5], **Pablo Bartolucci**[2,5], **Romain Arrestier**[1,2,3], **Ségolène Gendreau**[1,2,3], **Nicolas de Prost**[1,2,3], **Anne Hulin**[6], **Armand Mekontso Dessap**[1,2,3]

1 AP-HP, Hôpitaux Universitaires Henri-Mondor, Service de Médecine Intensive Réanimation, F-94010, Créteil, France, 2 Université Paris Est Créteil, INSERM, IMRB, Créteil, F-94010, France, 3 Université Paris Est Créteil, CARMAS, Créteil, F-94010, France, 4 Hôpital Intercommunal de Créteil, Service de Réanimation et Surveillance Continue Adulte, 94000, Créteil, France, 5 AP-HP, Hôpitaux Universitaires Henri-Mondor, Centre de Référence de la Drépanocytose, Créteil, France, 6 AP-HP, Hôpitaux Universitaires Henri Mondor, Service de Biochimie, Créteil, 94010 France

☉ These authors contributed equally to this work.
* keyvan.razazi@aphp.fr

**Data Availability Statement:** The datasets generated during and/or analyzed during the current study are not publicly available as consent

## Abstract

### Objective

Underdosing of antibiotics is common in patients with sickle cell disease (SCD). We hypothesized that in critically-ill patients with SCD receiving cefotaxime during acute chest syndrome, the continuous infusion may outperform the intermittent administration in achieving pharmacokinetic/pharmacodynamic targets.

### Design

Prospective before-after study.

### Settings

Intensive-care unit of a French teaching hospital and sickle cell disease referral center.

### Patients

Sixty consecutive episodes of severe acute chest syndrome in 58 adult patients with sickle cell disease.

### Interventions

Patients were treated with intermittent administration during the first period (April 2016 – April 2018) and with continuous infusion during the second period (May 2018 –August 2019).

for publication of raw data was not obtained from study participants. Data are available from the Mondor Institutional Data Access (contact via "ekeyvan.razazi@aphp.fr" or via the authors' institution at "rea.mondor@aphp.fr") for researchers who meet the criteria for access to confidential data.

**Funding:** This research did not receive any specific grant from funding agencies in the public, commercial, or not-for-profit sectors.

**Competing interests:** NO authors have competing interests.

## Measurements and main results

We included 60 episodes of acute chest syndrome in 58 patients (29 [25–34] years, 37/58 (64%) males). Daily dose of cefotaxime was similar between groups (59 [48–88] *vs.* 61 [57–64] mg/kg/day, p = 0.84). Most patients (>75%) presented a glomerular hyperfiltration with no difference between groups (p = 0.25). More patients had a cefotaxime trough level $\geq 2$ mg/L with continuous infusion than intermittent administration: 28 (93%) *vs.* 5 (16%), p<0.001. The median residual concentration was higher in the continuous infusion than intermittent administration group: 10.5 [7.4–13.3] *vs.* 0 [0–0] mg/L, p<0.001. No infection relapse was observed in the entire cohort. Hospital length of stay was similar between groups.

## Conclusion

As compared to intermittent administration, continuous infusion of cefotaxime maximizes the pharmacokinetic/pharmacodynamic parameters in patients with SCD. The clinical outcome did not differ between the two administration methods; however, the study was underpowered to detect such a difference.

## Introduction

Sickle-cell disease (SCD) is one of the most common severe monogenic disorders worldwide. Approximately, 300,000 infants are born annually worldwide with SCD and 100,000 people have SCD in the US [1]. Patients with SCD are at increased risk of severe bacterial infection, resulting primarily from reduced or absent splenic function. Acute chest syndrome (ACS) is a major complication of SCD and the first cause of mortality in adult patients [2]. ACS is characterized by fever and/or respiratory symptoms with new pulmonary infiltrates. Bacterial infection is documented in only a minority of cases [2,3]. However, empirical antibiotic therapy is often used in ACS because establishing a definitive etiology is not always possible. National guidelines and expert consensus strongly recommend antibiotic therapy for almost all patients with ACS despite the low quality of supporting evidence [1]. In clinical practice, empirical intravenous antimicrobial therapy is usually a combination therapy with a macrolide targeting intracellular bacteria and a beta lactam targeting pyogenic bacteria (amoxicillin in mild ACS and cephalosporin for severe ACS) [4]. Further studies are needed to improve the selection of patients requiring antibiotic therapy and/or the early stop of empiric antibiotic therapy in this setting as untreated infection may be fatal within hours in patients with SCD. ß-lactams are the most commonly prescribed antimicrobial agents in the critically-ill, especially in patients with acute lung injury [5]. ß-lactam antibiotics display time-dependent activity, where bacterial killing (microbiologic cure) and treatment efficacy (clinical cure) correlate with the duration of time that free (unbound) plasma drug concentrations remain above the minimal inhibitory concentration of the offending pathogen (*fT*>MIC). These characteristics suggest optimal effects with continuous infusion (CI) rather than traditional intermittent administration (IA) dosing. IA dosing may result in ß-lactam concentrations below the MIC for much of the dosing interval, increasing the risk of treatment failure and emergence of resistant strains. In patients with sepsis, interventions aimed at optimising antibiotic therapy demonstrated the greatest improvements in clinical outcomes [6,7].

An antibiotic concentration maintenance 4-fold higher than the minimal inhibitory concentration of the known or suspected pathogen is one of the main factors associated with bacterial killing. However, many critically ill patients with sepsis fall below this pharmacokinetic/pharmacodynamic target, in part because of augmented renal clearance. The use of continuous infusions (CI) improve the pharmacokinetic/pharmacodynamic targets in septic patients as compared to intermittent administration (IA) [8]. Underdosing of antibiotics is also common in patients with SCD [9,10], and associated with augmented renal clearance secondary to increased renal perfusion. In children with SCD, cefotaxime clearance increased by 22% during ACS [9]. Pharmacokinetic data on cefotaxime in adult patients with SCD are lacking.

We therefore hypothesized that in critically-ill patients with SCD receiving cefotaxime during ACS, the CI may outperform the IA in achieving pharmacokinetic/pharmacodynamic targets.

## Methods

### Design of the study

This prospective before-after study aimed at comparing cefotaxime administration by IA versus CI in consecutive adult patients admitted to our intensive-care unit for severe ACS between May 2016 and August 2019. The inclusion criterion was a severe ACS episode requiring ICU with cefotaxime administration. Severe ACS was defined as previously reported by ICU admission (as decided on the basis of a collegiate clinical assessment by the referring physician at the SCD clinic and the intensivist) based on one of the following signs [11]: respiratory rate greater than 30 breaths/minute, increased respiratory accessory muscle activity, partial pressure of oxygen in arterial blood less than 60 mm Hg on room air, respiratory acidosis, altered consciousness, extensive parenchymal opacities on the chest radiograph, and multiple organ failures. Exclusion criteria were the use of antibiotic regimen without cefotaxime, antibiotic dosage not performed, age <18 years old, and patient's refusal of data collected.

### ACS management

In this before-after study, consecutive patients were included without randomization. All patients were treated according to the French guidelines for the management of acute chest syndrome, with intravenous rehydration, multimodal analgesia (with controlled-release morphine if needed), preventive anticoagulation, oxygen supplementation, incentive spirometry, and red blood cell transfusion if indicated as per recommendations [11]. Antimicrobials were chosen as per European guidelines for the management of community-acquired pneumonia and French guidelines for the management of acute chest syndrome [4,12], suggesting the use of a third generation cephalosporine (e.g., cefotaxime), with or without association of an antibiotic against intracellular pathogen.

### Antibiotic administration

Thirty patients were treated with IA during the first period (April 2016–April 2018) and thirty with CI during the second period (May 2018–August 2019). Some results pertaining to the first period have been previously reported [10]. The target dose of cefotaxime was 60 mg/kg/day for 7 days (or less according to procalcitonin concentrations) in both groups. IA used 30–60 min intravenous perfusions q6-8 hours [1g q8 hours (n = 10); 2g q8 hours (n = 7); 1g q6 hours (n = 5); 1g q4 hours (n = 8)];, while CI used a 12-h infusion with an electric syringe pump, after a loading dose of 1g over 60 minutes. A loading dose of antibiotic before prolonged infusion is recommended to avoid delays in achieving effective beta-lactam

concentrations in critically-ill patients [13]. Before starting antibiotic treatment, we collected blood culture, sputum for gram stain and culture, and urine for *Legionella pneumophila* serogroup 1 and *Streptococcus pneumoniae* urinary antigen tests (BinaxNOW, Portland, ME, USA).

## Dosage

The plasma concentrations of cefotaxime were determined by a validated method using HPLC coupled with UV detection at 260 nm. Internal and external quality controls were regularly performed during the study period. Blood samples were obtained just before infusion of the drug (trough concentrations) and after at least the fourth dose (IA) or at any time after 24 hours (CI). The ß-lactam pharmacokinetic/pharmacodynamic target was considered achieved if the antibiotic concentration was 4-fold the minimal inhibitory concentration of cefotaxime for *Streptococcus pneumoniae* (i.e. 2mg/L).

## Definitions

Augmented renal clearance was defined as a an estimated glomerular filtration rate above 110 mL/min/1.73m$^2$, by using the CKD-EPI equation without adjustment for ethnicity, as previously suggested for SCD patients [14]. Infection relapse was defined as the growth of one or more of the initial causative bacterial strains (i.e., same genus, species) from a second sample taken from the same infection site 48 hours or more after stopping of antibiotics, combined with clinical signs or symptoms of infection. In our center, patients are systematically seen in consultation after hospital discharge following an episode of ACS. All data were collected from patient's chart and microbiology laboratory archives.

## Time and cost

The time and associated costs (drug and consumables) of administering a continuous versus an intermittent infusion were compared. One author (KR) observed 10 nurses administering, in succession, a continuous and an intermittent infusion in a cross-over design (five nurses began by the continuous infusion while five others began by the intermittent infusion). The time needed per day to prepare and administer the antibiotic was calculated for each infusion method (see online supplement). The costs for all items used in drug preparation and administration including syringe pump amortization expenses were also calculated (see online supplement). The cost of energy and labor cost of the nursing time were not considered.

## Ethics

The study was approved by the institutional Review Board of the French intensive care medicine society (CE SRLF 19–5) as a component of standard care and written and oral information about the study was given to the patients.

## Statistics

We estimated that a total of 30 consecutive episodes per group was deemed necessary in order to observe an increase of the percentage of patients with a trough concentration of cefotaxime ≥2mg/L from 17% with IA [10], to 60% [15,16] with CI, considering a power of 90%, and an alpha risk of 5%. Continuous data were expressed as median (25$^{th}$–75$^{th}$ percentiles) or mean ± standard deviation as appropriate and were compared using the Mann–Whitney test. Categorical variables, were evaluated using the chi square test or Fisher's exact test. The effect of CI on cefotaxime trough concentration ≥2 mg/L was also analyzed after adjustment on

estimated glomerular filtration rate. Correlations were tested using the Spearman's method. A two-tailed p value <0.05 was considered significant. Data were analyzed using IBM SPSS Statistics for Windows (version 19.0, IBM Corp Armonk, NY, USA).

## Results

### Patients

Sixty consecutive episodes of severe ACS (30 in each group) with a dosage of cefotaxime were assessed in 58 adult patients with SCD (see flow chart in online supplement). The demographics, baseline characteristics, clinical presentation, and laboratory values were similar between IA and CI groups (see Table 1), including age, weight, estimated glomerular filtration rate, and dose of cefotaxime received. The proportion of patients with augmented renal clearance at baseline and during ACS was similarly high (>75%) in both groups (Table 1). Liver function tests were not different between groups (Table 1). Lung bacterial infection was microbiologically proven in only one patient (third-generation cephalosporin-susceptible *Escherichia coli*).

### Cefotaxime concentration

Cefotaxime dosage were performed after a median of 3 days [1–4] The trough level concentration of cefotaxime was higher in the CI as compared to the IA group (10.5 [7.4–13.3] vs. 0 [0–0] mg/L, p <0.001) (Fig 1), despite similar median daily doses of cefotaxime (59 [48–88] vs. 61 [57–64] mg/kg/day, p = 0.84). A cefotaxime trough concentration ≥2 mg/L was more often observed in the CI group than in the IA group: 28 (93%) vs. 5 (17%), p<0.01. This result persisted after adjustment on estimated glomerular filtration: crude and adjusted odds ratio (95% confidence interval) of 70 (12–393), p<0.001 and 111 (15–818), p<0.001, respectively). There was no significant correlation between trough concentration and antibiotic daily dose (Spearman's Rho = 0.16, p = 0.84). One of the two patients with undetectable (<2 mg/L) trough concentrations in the CI group received an insufficient daily dose of cefotaxime (42 mg/kg). Episodes with undetectable (<2 mg/L) trough concentrations were more frequent in IA group (n = 25, 83%) than in CI group (n = 2; 7%), p<0.01.

### Outcome

No infection relapse was observed in the entire cohort. Three patients developed secondary ventilatory associated pneumonia (all in the IA group). We observed no adverse event associated with cefotaxime use, either neurologic (seizure, encephalopathy, coma, or delirium), hematological, or renal. Renal function at day 7 and day14 after antibiotic initiation did not differ between groups (Table 1). Only one patient died during the hospital stay, with no relation with an infectious process. Hospital length of stay was similar between groups.

### Time and associated costs

IA was performed 3 [3–6] times per day while CI was changed 2 [2–2] times per day. Daily nurse timework was lower with CI as compared to IA: 234 [234–234] vs 278 [278–556] seconds, p<0.001. Cost of all necessary material and drug were lower with CI as compared to IA: 6.1 [6.1–7.0] vs 7.8 [5.6–11.1] euros, p<0.01.

## Discussion

To the best of our knowledge, this is the first study to report improved pharmacokinetic/pharmacodynamic targets with CI (as compared to IA) of ß-lactams in SCD patients with ACS.

**Table 1. Baseline characteristics of 60 patients with acute chest syndrome according to cefotaxime administration.**

| Parameter | Intermittent administration (n = 30) | Continuous infusion (n = 30) | P value |
|---|---|---|---|
| Age, years | 30.4 [24.8–36.8] | 28.3 [25.4–33.4] | 0.34 |
| Male gender | 19 (63%) | 19 (63%) | 0.99 |
| Weight, kg | 70 [60.5–79.8] | 69.5 [60.3–75.0] | 0.57 |
| Height, cm | 174 [170–175] | 175 [170–183] | 0.56 |
| Baseline total Hb, g/dL | 8.5 [8.0–9.5] | 8.5 [8.0–9.0] | 0.92 |
| SS genotype | 26 (86%) | 28 (93%) | 0.55 |
| Past medical history: | | | |
| Previous ACS | 21 (70%) | 27 (90%) | 0.11 |
| Previous VOC | 27 (90%) | 28 (93%) | 0.99 |
| Stroke | 1 (3%) | 3 (10%) | 0.61 |
| Retinopathy | 5 (17%) | 7 (23%) | 0.75 |
| Priapism | 3 (10%) | 4 (13%) | 0.99 |
| Bone necrosis | 10 (33%) | 8 (27%) | 0.78 |
| Heart disease | 6 (20%) | 5 (17%) | 0.99 |
| Alloimmunization | 3 (10%) | 8 (27%) | 0.18 |
| Baseline creatinine (μmol/L) | 56 [48–64] | 57 [49–62] | 0.98 |
| Baseline eGFR, mL/min/1.73m$^2$ | 130 [124–135] | 129 [119–139] | 0.94 |
| Baseline augmented renal clearance | 26 (87%) | 27 (90%) | 0.99 |
| Baseline treatments | | | |
| Transfusion therapy | 1 (3%) | 5 (17%) | 0.19 |
| Hydroxyurea treatment | 16 (53%) | 17 (57%) | 0.99 |
| Home oxygen therapy | 3 (10%) | 6 (20%) | 0.47 |
| Type of precipitating factor | | | 0.81 |
| VOC before ACS | 25 (83%) | 24 (80%) | |
| Surgery | 2 (7%) | 1 (3%) | |
| Infection | 2 (7%) | 1 (3%) | |
| Other | 1 (3%) | 4 (13%) | |
| Symptoms | | | |
| Temperature,˚C | 38.3 [37.7–39] | 37.5 [36.9–38.5] | 0.046* |
| Chest pain | 26 (87%) | 24 (80%) | 0.73 |
| Extrathoracic pain during ACS | 23 (77%) | 21 (70%) | 0.77 |
| Cough | 10 (33%) | 7 (23%) | 0.57 |
| Pulmonary crackles | 22 (73%) | 18 (60%) | 0.41 |
| Jugular venous distension | 0 (0%) | 2 (7%) | 0.49 |
| Heart rate, beats/min | 104 [100–116] | 99 [90–122] | 0.45 |
| Respiratory rate, breaths/min | 26 [22 – 36] | 25 [19 – 31] | 0.38 |
| Oxygen therapy, L/min | 4 [2 – 6] | 4 [3 – 5] | 0.41 |
| Pulse oxygen saturation, % | 97 [96–99] | 98 [96–99] | 0.38 |
| Acute cor pulmonale | 5 (17%) | 4 (13%) | 0.99 |
| Laboratory values at ACS diagnosis | | | |
| White cell count, $10^9$/L | 17.8 [13.9–23.7] | 17.3 [13.5–22.2] | 0.6 |
| Platelet count,$10^9$/L | 268 [194–344] | 320 [244–386] | 0.16 |
| Total hemoglobin, g/dL | 8.0 [6.2–9.5] | 7.9 [6.6–8.8] | 0.67 |
| Lactate dehydrogenase, IU/L | 619 [151–1254] | 560 [86–713] | 0.17 |
| Aspartate aminotransferase, IU/L | 59 [38–97] | 59 [43–102] | 0.91 |
| Alanine aminotransferase (IU/L) | 35 [22–51] | 24 [18–58] | 0.42 |

*(Continued)*

**Table 1.** (Continued)

| Parameter | Intermittent administration (n = 30) | Continuous infusion (n = 30) | P value |
|---|---|---|---|
| Total bilirubine (μmol/L) | 59 [37–77] | 52 [27–77] | 0.38 |
| Prothrombin index, % | 67 [53–72] | 69 [59–75] | 0.41 |
| Creatinine (μmol/L) # | 47 [31–61.8] | 56 [45–61] | 0.1 |
| eGFR, mL/min/1.73m$^2$# | 144 [126–154] | 131 [124–140] | 0.17 |
| Augmented renal clearance | 24 (80%) | 28 (93%) | 0.25 |
| ACS associated treatments | | | |
| Hyperhydratation | 30 (100%) | 30 (100%) | 0.99 |
| Exchange and single transfusion | 22 (73%) | 19 (63%) | 0.41 |
| Antibiotics | | | |
| Time from cefotaxime initiation to antibiotic dosage (days) | 4 [3–4] | 3 [1–4] | 0.19 |
| Dose of cefotaxime received, mg/kg/day | 59.1 [47.6–87.6] | 60.6 [57.3–63.8] | 0.84 |
| Plasma cefotaxime concentration, mg/dL | 0 [0–0] | 10.5 [7.4–13.3] | < 0.001* |
| Outcome | | | |
| Death | 1 (3%) | 0 | 0.99 |
| Mechanical ventilation | 4 (13%) | 1 (3%) | 0.35 |
| Non-invasive ventilation | 0 (0%) | 1 (3%) | 0.32 |
| Length of hospital stay (days) | 13.5 [12–16.75] | 13.5 [9.5–17] | 0.81 |
| Creatinine (μmol/L) at day 7 | 45 [35–58] | 50 [42–61] | 0.19 |
| Creatinine (μmol/L) at day 14 | 57 [44–68] | 57 [52–72] | 0.59 |

Data are presented as median [25th-75th percentiles] or number (percentage); Hb, hemoglobin; ACS, acute chest syndrome; VOC, vaso-occlusive pain crisis; SS genotype: Homozygous sickle cell (SS) disease; alloimmunization: Immunization with at least one significant antibody [17].

# the day of antibiotic dosage, eGFR = estimated glomerular filtration rate using the CKD-EPI equation without adjustment for ethnicity (Arlet *et al*, 2012), Acute cor pulmonale was defined as dilated right ventricle (end-diastolic right ventricle / left ventricle area ratio >0.6) associated with septal dyskinesia

* p < 0.05.

In previous studies in SCD patients, the incidence of documented bacterial infections was low, which is consistent with our results. This low incidence of bacterial infection cannot be used to totally question the liberal use of antibiotics in case of ACS with fever [1], although procalcitonin may help shorten the antibiotic course [18].

SCD complicated by an ACS exposes to augmented renal clearance, leading to a high risk of underdosing treatment, especially ß-lactams. In patients with SCD, there is a potential higher risk of secondary infections, especially, bone-joint infection complicating bloodstream infections. The pharmacokinetic/pharmacodynamic index associated with optimal ß-lactam activity is the percent fraction of time above the minimal inhibitory concentration. Data in critically ill patients suggest that they may benefit from longer (e.g. fraction of time above the minimal inhibitory concentration) [15] and higher (e.g. 2–5 times the minimal inhibitory concentration) ß-lactam exposures with CI. The loading dose performed to achieve effective beta-lactam concentrations could not explain on its own the difference of concentration between groups as dosage were performed 3 days after cefotaxime initiation. The results of the DALI study support the conclusions that better outcomes for critically ill patients can be expected with higher drug exposures [15]. Prolonged infusion schemes increase the fraction of the dose interval in which unbound antibiotic concentrations exceed MIC of the pathogen ($f$T > MIC) compared with standard intermittent infusion [19–21]. A recent meta-analysis showed that hospital mortality was significantly lower and clinical cure was significantly higher in the CI group than in the IA group [7]. All but two patients in the CI group had a cefotaxime trough

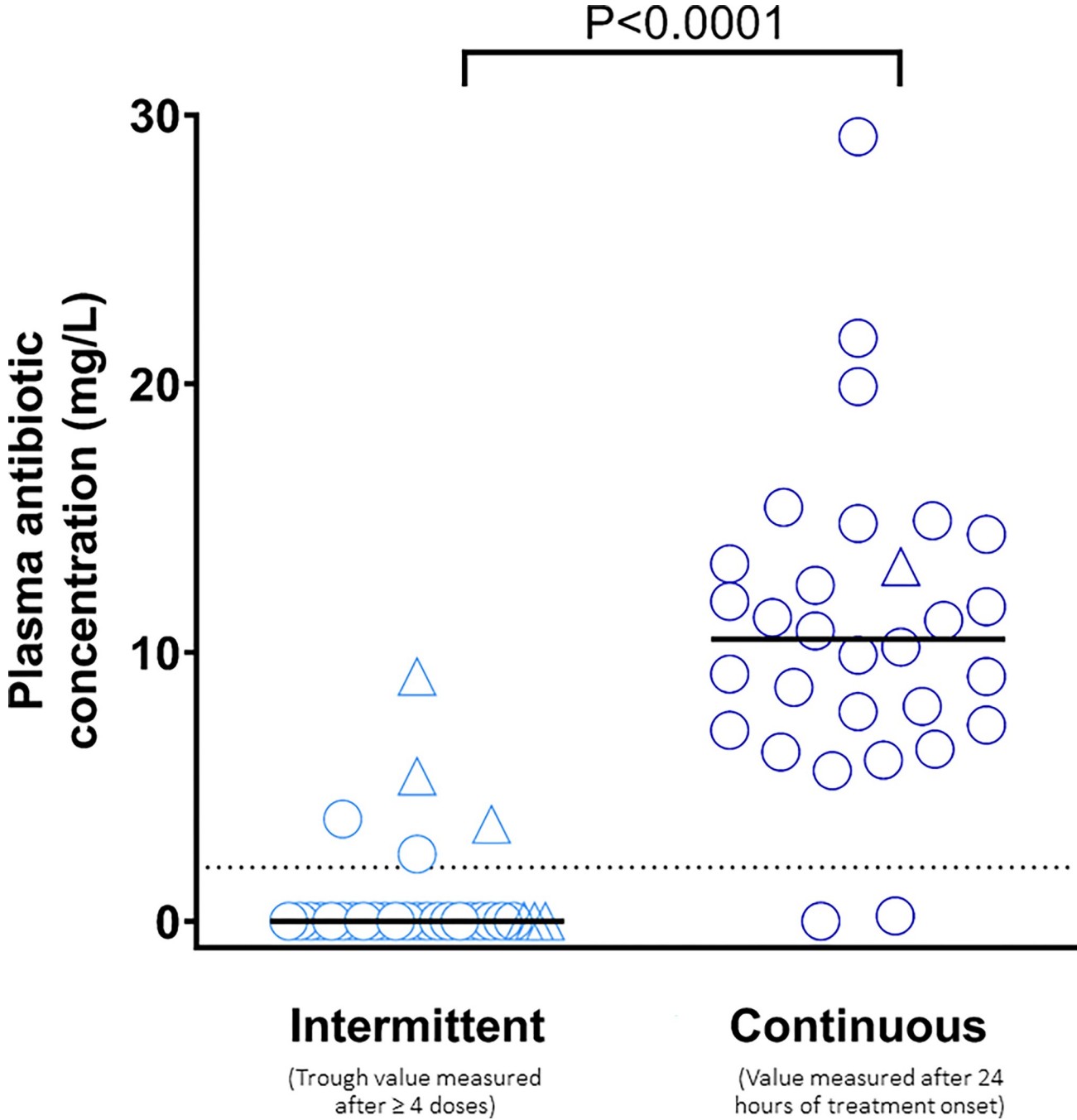

**Fig 1. Plasma antibiotics concentration according to type of antibiotic administration.** Circles and triangles denote patients with and without augmented renal clearance, respectively.

level concentration $\geq 2$ mg/L, while it was undetectable ($<2$ mg/L) in all but five patients of the IA group. Our findings have major clinical implications, given the burden of severe infections and difficult to treat infections in patients with SCD[22,23]. Our results are in accordance with those of Maksoud et al who found very low attainment of target cefotaxime serum concentrations in pediatric SCD patients (only 5.6% of patients achieved 90% of time above the MIC of 2mg/L) [24]. Our results suggest promoting CI of time dependent antibiotics instead of bolus

IA during ACS. This strategy seems safe to optimize pharmacokinetic/pharmacodynamic parameters allowing a plasmatic concentration at least 4 times above the minimal inhibitory concentration of usual bacteria associated with ACS. Moreover, nurse timework and costs were decreased with CI as previously published with other antibiotics [25]. The time saved for nurses was due to the reduction in the number of administrations per day. However, this result was obtained in a simulation room, and the difference was numerically marginal (less than one minute). On the other hand, CI requires syringe pump that may not always be available in low-income countries.

Our study has several limitations, including the small sample size, the assessment of a single antibiotic at a given dose, and the lack of clinical outcome. In addition, the before-after design and monocenter setting in an intensive care unit may infer several bias. Most patients presented with augmented renal clearance, a frequent feature in patients with SCD at baseline and during ACS. Our results cannot be extrapolated to SCD patients with decreased renal function, to other antibiotics with a concentration-dependent therapeutic effect such as aminoglycosides or to patients without SCD. Further studies are needed to assess the feasibility of this approach in medical wards. Further studies are also warranted to assess the effect of the optimization of pharmacokinetic/pharmacodynamic parameters on clinical outcomes in this setting, inasmuch as the rate of bacterial documentation is low during ACS. Our findings suggests exploring the pharmacokinetic/pharmacodynamic of other antibiotics (especially for bloodstream and difficult to treat infections like bone-joint infections [26]) and that of other treatments influenced by renal clearance (e.g., morphine) in patients with SCD.

## Conclusion

As compared to IA, CI of cefotaxime maximizes the pharmacokinetic/pharmacodynamic parameters. The median residual concentration was higher in the CI than IA group. CI of time-dependent antibiotics seems to decrease the risk of underdosing in patients with SCD. Daily nurse timework and cost were also lower with CI. The clinical outcome did not differ between the two administration methods; however, the study was underpowered to detect such a difference. Further studies are needed to assess the impact of CI on clinical outcomes in patients with SCD.

## Author Contributions

**Conceptualization:** Keyvan Razazi, Enora Berti, Jerome Cecchini, Anoosha Habibi, Pablo Bartolucci, Romain Arrestier, Nicolas de Prost, Armand Mekontso Dessap.

**Data curation:** Keyvan Razazi, Enora Berti, Jerome Cecchini, Guillaume Carteaux, Anoosha Habibi, Romain Arrestier.

**Investigation:** Guillaume Carteaux.

**Methodology:** Keyvan Razazi, Enora Berti, Jerome Cecchini, Pablo Bartolucci, Romain Arrestier, Ségolène Gendreau, Nicolas de Prost, Anne Hulin, Armand Mekontso Dessap.

**Project administration:** Armand Mekontso Dessap.

**Resources:** Anoosha Habibi.

**Supervision:** Keyvan Razazi.

**Validation:** Keyvan Razazi, Anne Hulin, Armand Mekontso Dessap.

**Writing – original draft:** Keyvan Razazi, Enora Berti, Armand Mekontso Dessap.

**Writing – review & editing:** Keyvan Razazi, Enora Berti, Jerome Cecchini, Guillaume Car-
teaux, Anoosha Habibi, Pablo Bartolucci, Romain Arrestier, Ségolène Gendreau, Nicolas de
Prost, Anne Hulin, Armand Mekontso Dessap.

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
