## [Decision Letter · Decision Letter 0]

28 Jun 2022

PONE-D-21-21735

Decreased risk of underdosing with continuous infusion versus intermittent administration of Cefotaxime in patients with sickle cell disease and acute chest syndrome.

PLOS ONE

Dear Dr. Razazi,

Thank you for submitting your manuscript to PLOS ONE. Firstly, we would like to apologize for the delay in processing your manuscript. It has been exceptionally difficult to secure reviewers to evaluate your study. We have now received one completed review, which is available below. The reviewer has raised significant scientific concerns about the study that need to be addressed in a revision.

Please note that we have only been able to secure a single reviewer to assess your manuscript. We are issuing a decision on your manuscript at this point to prevent further delays in the evaluation of your manuscript. Please be aware that the editor who handles your revised manuscript might find it necessary to invite additional reviewers to assess this work once the revised manuscript is submitted. However, we will aim to proceed on the basis of this single review if possible.

After careful consideration, we feel that it has merit but does not fully meet PLOS ONE’s publication criteria as it currently stands. Therefore, we invite you to submit a revised version of the manuscript that addresses the points raised during the review process.

We look forward to receiving your revised manuscript.

Kind regards,

Miquel Vall-llosera Camps

Senior Editor

PLOS ONE

Journal Requirements:

2. Please provide additional details regarding participant consent. In the Methods section, please ensure that you have specified (1) whether consent was informed and (2) what type you obtained (for instance, written or verbal). If your study included minors, state whether you obtained consent from parents or guardians. If the need for consent was waived by the ethics committee, please include this information.

3. Thank you for stating the following financial disclosure: "Unfunded studies"

4. Thank you for stating the following in your Competing Interests section: "NO authors have competing interests"

Reviewers' comments:

Reviewer's Responses to Questions

**Comments to the Author**

1. Is the manuscript technically sound, and do the data support the conclusions?

Reviewer #1: Yes

2. Has the statistical analysis been performed appropriately and rigorously? 

Reviewer #1: Yes

3. Have the authors made all data underlying the findings in their manuscript fully available?

Reviewer #1: No

4. Is the manuscript presented in an intelligible fashion and written in standard English?

Reviewer #1: Yes

5. Review Comments to the Author

Reviewer #1: The authors have conducted a prospective study comparing intermittent administration of cefoxatime versus continuous infusion for in adults diagnosed with acute chest syndrome (ACS). This is a nearly unique study about the administration of antibiotics in a critically important complication of sickle cell disease. Acute chest syndrome is one of the leading causes of death in hospitalized patients with sickle cell. Sickle cell disease has historically been hard to study, with most studies failing because of lack of enrollment. So any good data is important to get out into the accessible literature. The conclusion that a mainstay of treatment for a life-threatening complication may not be effective because of the delivery mention is important, interesting, and relevant. Even though many centers don’t use cefoxatime as their first-line antibiotic, the same concerns of hyperfiltration and pulse versus continuous dosing requires those centers to consider if their protocols may need adjusting. The authors are commended for a clean study, with an appropriate discussion of limitations.

In short, this study, though limited, is an important addition to prior work that may be exposing a serious flaw in the treatment of a life-threatening condition.

6. PLOS authors have the option to publish the peer review history of their article (what does this mean?). If published, this will include your full peer review and any attached files.

Reviewer #1: No

---

## [Author Response · Author response to Decision Letter 0]

8 Sep 2022

The authors have conducted a prospective study comparing intermittent administration of cefoxatime versus continuous infusion for in adults diagnosed with acute chest syndrome (ACS). This is a nearly unique study about the administration of antibiotics in a critically important complication of sickle cell disease. Acute chest syndrome is one of the leading causes of death in hospitalized patients with sickle cell. Sickle cell disease has historically been hard to study, with most studies failing because of lack of enrollment. So any good data is important to get out into the accessible literature. The conclusion that a mainstay of treatment for a life-threatening complication may not be effective because of the delivery mention is important, interesting, and relevant. Even though many centers don’t use cefotaxime as their first-line antibiotic, the same concerns of hyperfiltration and pulse versus continuous dosing requires those centers to consider if their protocols may need adjusting. The authors are commended for a clean study, with an appropriate discussion of limitations.

In short, this study, though limited, is an important addition to prior work that may be exposing a serious flaw in the treatment of a life-threatening condition.

We sincerely thank the reviewer for their constructive comments. All comments have been addressed and changes made in the manuscript. We have submitted a point by point response to all reviewers comments. We feel those changes improved the manuscript. 

Sincerely

Major comments

1. The acknowledged key limitation is the difference between statistical and clinical significance. Even though the intermittent administration dose had lower troughs, there was no clinical difference. Since not all ACS has bacterial involvement, it likely would take more outcomes to separate a difference. The authors make a compelling case for further studies. 

Response: We agree with reviewer’s comment. In accordance, we added a sentence in the limitation section of the discussion as follows: Further studies are also warranted to assess the effect of the optimization of pharmacokinetic/pharmacodynamic parameters on clinical outcomes in this setting, inasmuch as the rate of bacterial documentation is low during ACS.

2. The study design implied that only the first thirty patients from each study period were enrolled. Where there patients with ACS not enrolled during these periods. If not, why not. The more data the better. Also were there any repeat patients that got both intermittent and continuous infusions of cefotaxime? Table 1 shows a high percentage of “Previous ACS.” If there were overlap, a brief separate paired analysis would be of interest.

Responses: There is no overlap between periods. Only two patients got the two infusion methods, during two separate ACS episodes, precluding the use of a paired analysis in this subgroup. Dosage results for those two patients were zero in intermittent infusion period and over the 2 mg/L cutoff (21.7 and 6.3 mg/l) in the continuous infusion period. In accordance with reviewer’s comment, a flow chart was provided in the manuscript supplement and the result section was modified as follows: “Sixty consecutive episodes of severe acute chest syndrome (30 in each group) with a dosage of cefotaxime were assessed in 58 adult patients with sickle cell disease (see flow chart in online supplement)”. 

3. The timing of the trough lab values is critical. Was there an enforced standard to ensure the troughs were drawn within an acceptable window.

Response: As mentioned in the methods section, for the intermittent administration period, blood samples were obtained just before infusion of the drug (trough concentrations). We agree with reviewer that a dosage sampled not just before the infusion could falsely increase the concentration, but in this period most concentrations (n=25; 85%), were <2mg/L . As mentioned in the methods section, for the continuous infusion period, blood samples were obtained after 24 hours of infusion. 

4. The range of cefotaxime trough level concentrations in the intermittent administration group was 0 - 0 mg/l, then later reported that 17% were in AI was over the 2 mg/L cutoff. This was confusing.

Response: 0 - 0 mg/l refers to the interquartile range. More than 75% of patients (n=25, 85%) had a concentration of 0, explaining why the 1st and 3rd quartile were 0. 

5. Lastly, if the authors had any information on the cost comparison of the two approaches, this would be invaluable. If the cost, time, and labor of the intermittent administration exceeded the continuous administration, then a compelling case for continuous administration could be made.

Response: The time and associated costs (drug and supply) of administering a continuous versus an intermittent infusion were compared. We observed 10 nurses for each infusion method. The time needed per day to prepare and administer the antibiotic was calculated for each infusion method. Daily nurse timework was lower with continuous infusion as compared to bolus infusion : 234 [234-234] vs 278 [278-556] seconds, p< 0.001. The costs for all items used in drug preparation and administration including syringe pump amortization expenses were also calculated. Cost of all necessary material and drug were lower with continuous infusion as compared to bolus infusion: 6.1 [6.0-6.1] vs 8.2 [7.9-15.9] euros, p< 0.001. Labor cost of the nursing time was not considered.

These results are in accordance with previous study on cost analysis of continuous infusion of other antibiotics [1]. However, we should also consider the limited availability of syringe pump in some low-income countries. These points were added in the manuscript (methods, results, and discussion). 

Minor comments: 

1. Generic drug names are not capitalized.

Responses: Cefotaxime was changed for cefotaxime 

2. Cefotaxime is not the first-line antibiotics in many centers and when used it is often at a higher dose (60 to 100 mg/kg/day). The limitations (line 190) could possibly add something like “assessment of a single antibiotic at a single dose”

Response: in accordance with reviewer comment, we added the following sentence as a limitation: “Our study has several limitations, including the assessment of a single antibiotic at a given dose”. We also added the following data in the result section: “There was no significant correlation between trough concentration and antibiotic daily dose (Spearman’s Rho = 0.16, p=0.84).”

3. Given the n of 30, I’m not sure why a non-parametric test was used. Usually if the n is greater than twenty, more standard parametric tests (t-tests, ANOVA) can be used and are easier to interpret. Although not perfect, several simple tests for normality ( Wilkes, e.g.) can by use to test if the distribution is “normal enough. The authors may want to analyze their data with parametric statistics to see if they get the same conclusion. This would make their conclusion more robust. 

Response: As many concentration were 0 in the intermittent infusion group, the distribution was not normal, with a p value for Kolmogorov-Smirnov test < 0.001

4. I wasn’t sure why the categorical variables were changed to a percentage before Chi-2 testing. Usually the raw counts are tested. Any mathematical manipulation changes the information in the data. Again, they may want to perform the Chi-2 testing on the raw data for robustness.

Response: Sorry if the sentence was not understandable. Categorical variables were not transformed; we corrected this point in the statistical section. 

1. Florea NR, Kotapati S, Kuti JL, Geissler EC, Nightingale CH, Nicolau DP. Cost analysis of continuous versus intermittent infusion of piperacillin-tazobactam: a time-motion study. Am J Health-Syst Pharm AJHP Off J Am Soc Health-Syst Pharm. 2003;60:2321–7.

---

## [Decision Letter · Decision Letter 1]

2 Dec 2022

PONE-D-21-21735R1

Decreased risk of underdosing with continuous infusion versus intermittent administration of Cefotaxime in patients with sickle cell disease and acute chest syndrome.

PLOS ONE

Dear Dr. Razazi,

Thank you for submitting your manuscript to PLOS ONE. After careful consideration, we feel that it has merit but does not fully meet PLOS ONE’s publication criteria as it currently stands. Therefore, we invite you to submit a revised version of the manuscript that addresses the points raised during the review process.

The paper has been reassessed by reviewer #1 and additional reviewers have been solicited to provide an assessment. Reviewer #1 has some additional minor comments which we would like you to address. 

We look forward to receiving your revised manuscript.

Kind regards,

Academic Editor

PLOS ONE

Journal Requirements:

Reviewers' comments:

Reviewer's Responses to Questions

**Comments to the Author**

1. If the authors have adequately addressed your comments raised in a previous round of review and you feel that this manuscript is now acceptable for publication, you may indicate that here to bypass the “Comments to the Author” section, enter your conflict of interest statement in the “Confidential to Editor” section, and submit your "Accept" recommendation.

Reviewer #1: All comments have been addressed

Reviewer #2: All comments have been addressed

Reviewer #3: All comments have been addressed

Reviewer #4: All comments have been addressed

2. Is the manuscript technically sound, and do the data support the conclusions?

Reviewer #1: Yes

Reviewer #2: Partly

Reviewer #3: Yes

Reviewer #4: Yes

3. Has the statistical analysis been performed appropriately and rigorously? 

Reviewer #1: Yes

Reviewer #2: Yes

Reviewer #3: Yes

Reviewer #4: Yes

4. Have the authors made all data underlying the findings in their manuscript fully available?

Reviewer #1: Yes

Reviewer #2: Yes

Reviewer #3: Yes

Reviewer #4: Yes

5. Is the manuscript presented in an intelligible fashion and written in standard English?

Reviewer #1: Yes

Reviewer #2: Yes

Reviewer #3: Yes

Reviewer #4: Yes

6. Review Comments to the Author

Reviewer #1: The authors were diligent in address all of the reviewers' comments. The manuscript reads much more smoothly now. The combination of the pharmacokinetics and cost analyses in augmented renal clearance were compelling. I only have four minor editing suggestions. 1. In table one you list cor pulmonale. Since there are a few different definitions, you may consider a footnote of how it was defined here. 2. Figure 1. You may consider adding a line under Intermittent to say it would a trough v after >= the fourth dose alue and a line under continuous to say it was after 24 hours. That way the figure is free standing. 3. In the discussion, you state a higher risk of secondary infections with underdosing. You could either state that there is a potential higher risk of secondary infections or add a cite. 4. Lastly, a line in the paragraph may read better if you state Most patients presented with augmented renal clearance, adding the with.

Great job.

Reviewer #2: The authors had responded vigorously the reviewer's comments, but some major limitations still.

1. The difference between statistical and clinical significance.

2. There is different classification in pharmacodynamics of antibiotics, such as time-dependent or concentration-dependent. studies in one antibiotics seems not to be explored to other situation of medical clinic.

Reviewer #3: The authors have addressed the comments raised by the reviewer appropriately. The manuscript may be permitted for the publication.

Reviewer #4: The authors of the study "Decreased risk of underdosing with continuous infusion versus intermittent administration of Cefotaxime in patients with sickle cell disease and acute chest syndrome“ addressed all comments of the previous reviewers. I have nothing more to add.

7. PLOS authors have the option to publish the peer review history of their article (what does this mean?). If published, this will include your full peer review and any attached files.

Reviewer #1: No

Reviewer #2: No

Reviewer #3: **Yes: **Pachamuthu Balakrishnan

Reviewer #4: No

---

## [Author Response · Author response to Decision Letter 1]

15 Dec 2022

We sincerely thank the reviewer for their constructive comments. All comments have been addressed and changes made in the manuscript. We have submitted a point-by-point response to all reviewers’ comments. 

Sincerely

Reviewer #1: The authors were diligent in address all of the reviewers' comments. The manuscript reads much more smoothly now. The combination of the pharmacokinetics and cost analyses in augmented renal clearance were compelling. I only have four minor editing suggestions. 1. In table one you list cor pulmonale. Since there are a few different definitions, you may consider a footnote of how it was defined here. 

Response: in accordance with reviewer comment, a footnote was added in table 1. 

2. Figure 1. You may consider adding a line under Intermittent to say it would a trough v after >= the fourth dose alue and a line under continuous to say it was after 24 hours. That way the figure is free standing. 

Response: in accordance with reviewer comment, figure 1 was modified.

3. In the discussion, you state a higher risk of secondary infections with underdosing. You could either state that there is a potential higher risk of secondary infections or add a cite. 

Response: in accordance with reviewer comment, the sentence was changed with “potential”

4. Lastly, a line in the paragraph may read better if you state Most patients presented with augmented renal clearance, adding the with.

Response: in accordance with reviewer comment, “with” was added.

Great job.

Thank you 

Reviewer #2: The authors had responded vigorously the reviewer's comments, but some major limitations still.

1. The difference between statistical and clinical significance.

Response: we completely agree with reviewer comment, a comment was added in the limitation section concerning the lack of clinical outcome

2. There is different classification in pharmacodynamics of antibiotics, such as time-dependent or concentration-dependent. studies in one antibiotics seems not to be explored to other situation of medical clinic.

Response: A sentence was added in the limitation section concerning the fact that our results cannot be extrapolated to other antibiotics with a concentration-dependent therapeutic effect such as aminoglycosides

Reviewer #3: The authors have addressed the comments raised by the reviewer appropriately. The manuscript may be permitted for the publication.

Response : thank you 

Reviewer #4: The authors of the study "Decreased risk of underdosing with continuous infusion versus intermittent administration of Cefotaxime in patients with sickle cell disease and acute chest syndrome“ addressed all comments of the previous reviewers. I have nothing more to add

Response : thank you

---

## [Decision Letter · Decision Letter 2]

11 Jan 2023

PONE-D-21-21735R2Decreased risk of underdosing with continuous infusion versus intermittent administration of Cefotaxime in patients with sickle cell disease and acute chest syndrome.PLOS ONE

Dear Dr. Razazi,

Thank you for submitting your manuscript to PLOS ONE. After careful consideration, we feel that it has merit but does not fully meet PLOS ONE’s publication criteria as it currently stands. Therefore, we invite you to submit a revised version of the manuscript that addresses the points raised during the review process.

Please revise.

We look forward to receiving your revised manuscript.

Kind regards,

Academic Editor

PLOS ONE

Journal Requirements:

Reviewers' comments:

Reviewer's Responses to Questions

**Comments to the Author**

1. If the authors have adequately addressed your comments raised in a previous round of review and you feel that this manuscript is now acceptable for publication, you may indicate that here to bypass the “Comments to the Author” section, enter your conflict of interest statement in the “Confidential to Editor” section, and submit your "Accept" recommendation.

Reviewer #2: All comments have been addressed

Reviewer #4: All comments have been addressed

2. Is the manuscript technically sound, and do the data support the conclusions?

Reviewer #2: Yes

Reviewer #4: Yes

3. Has the statistical analysis been performed appropriately and rigorously? 

Reviewer #2: Yes

Reviewer #4: Yes

4. Have the authors made all data underlying the findings in their manuscript fully available?

Reviewer #2: Yes

Reviewer #4: Yes

5. Is the manuscript presented in an intelligible fashion and written in standard English?

Reviewer #2: Yes

Reviewer #4: Yes

6. Review Comments to the Author

Reviewer #2: The authors have adequately addressed my comments raised in a previous round of review. I do not have any more question.

Reviewer #4: All comments of the previous reviews have been addressed. Some formal suggestions:

There was one phrase that I did not understand:

What do the authors mean by the phrase “Eight internal and external quality

controls were regularly performed“. Over what time period?

Some formal issues

- please check spelling of affiliations

- check spelling in the text always write cefotaxime starting with a capital letter or small letter

- Use SI units (h for hour), not hr

- please check for blank spaces in the text and punctuatio

- when abbreviations are introduced in the text, they should be used throughout the whole document

- rather write chi-square test than Chi 2 test

7. PLOS authors have the option to publish the peer review history of their article (what does this mean?). If published, this will include your full peer review and any attached files.

Reviewer #2: No

Reviewer #4: No

---

## [Author Response · Author response to Decision Letter 2]

27 Jan 2023

We sincerely thank the reviewer for their constructive comments. All comments have been addressed and changes made in the manuscript. We have submitted a point-by-point response to all reviewers’ comments. 

All authors have read and approved the submission of the manuscript, and the manuscript has not been published and is not being considered for publication elsewhere in whole or in part in any language. Please, feel free to contact me for any requirements related to this submission.

Thank you so much for your attention.

Sincerely,

Keyvan Razazi, MD

Corresponding Author

Service de Réanimation Médicale, Hôpital Henri Mondor, Créteil, France

E-mail : keyvan.razazi@aphp.fr

Sincerely

Reviewer #2: The authors have adequately addressed my comments raised in a previous round of review. I do not have any more question.

Reviewer #4: All comments of the previous reviews have been addressed. Some formal suggestions:

There was one phrase that I did not understand:

What do the authors mean by the phrase “Eight internal and external quality

controls were regularly performed“. Over what time period?

Response: Internal quality controls are used to validate the assay series while external quality controls allow our assays to be compared to our peers.

“Eight Internal and external quality controls were regularly performed during the study period.

Some formal issues

- please check spelling of affiliations

- check spelling in the text always write cefotaxime starting with a capital letter or small letter

- Use SI units (h for hour), not hr

- please check for blank spaces in the text and punctuatio

- when abbreviations are introduced in the text, they should be used throughout the whole document

- rather write chi-square test than Chi 2 test

Response: In accordance with reviewer comment, all changes have been made.

---

## [Decision Letter · Decision Letter 3]

7 Feb 2023

PONE-D-21-21735R3Decreased risk of underdosing with continuous infusion versus intermittent administration of Cefotaxime in patients with sickle cell disease and acute chest syndrome.PLOS ONE

Dear Dr. Razazi,

Thank you for submitting your manuscript to PLOS ONE. After careful consideration, we feel that it has merit but does not fully meet PLOS ONE’s publication criteria as it currently stands. Therefore, we invite you to submit a revised version of the manuscript that addresses the points raised during the review process.

Please revise.

We look forward to receiving your revised manuscript.

Kind regards,

Academic Editor

PLOS ONE

Reviewers' comments:

Reviewer's Responses to Questions

**Comments to the Author**

1. If the authors have adequately addressed your comments raised in a previous round of review and you feel that this manuscript is now acceptable for publication, you may indicate that here to bypass the “Comments to the Author” section, enter your conflict of interest statement in the “Confidential to Editor” section, and submit your "Accept" recommendation.

Reviewer #4: All comments have been addressed

Reviewer #5: (No Response)

2. Is the manuscript technically sound, and do the data support the conclusions?

Reviewer #4: Yes

Reviewer #5: Partly

3. Has the statistical analysis been performed appropriately and rigorously? 

Reviewer #4: Yes

Reviewer #5: No

4. Have the authors made all data underlying the findings in their manuscript fully available?

Reviewer #4: Yes

Reviewer #5: Yes

5. Is the manuscript presented in an intelligible fashion and written in standard English?

Reviewer #4: Yes

Reviewer #5: Yes

6. Review Comments to the Author

Reviewer #4: The authors of the manuscript "Decreased risk of underdosing with continuous infusion versus intermittent administration of cefotaxime in patients with sickle cell disease and acute chest syndrome.“ addressed all suggestions of the previous reviews. I congratulate the authors for this interesting research. I have nothing more to add.

Reviewer #5: This is a good paper that provided new insight and merit advancing to the next stage. However given the paucity of the data presented, I will suggest that it is best considered as a research letter rather than a full original article. See my few improvement suggestions below:

Introduction:

• Line 64-73: The introduction could provide more background on the epidemiology of SCD and ACS and why these conditions are important to study.

• Add a brief overview of previous studies investigating the use of Cefotaxime in SCD and ACS.

• Discuss the current state of empirical antibiotic therapy in ACS and the need for further research in this area.

Methods:

• Generally, the STROBE checklist for observational study should be followed for the manuscript including the method section

• Line 89-92: Consider specifying the inclusion and exclusion criteria for the study population, such as the criteria for being admitted to the ICU for severe ACS and age range of patients.

• Line 105-106: The methods for collecting data on hospital length of stay and infection relapse could be described in more detail.

• Consider adding a power analysis to estimate the sample size necessary for the study.

• Discuss the assumptions made when carrying out the statistical analysis.

• Line 98-100: The methods for measuring Cefotaxime trough levels and residual concentrations could be described in more detail.

• Line 89-93: The design of the study could be described in more detail, including the inclusion and exclusion criteria for patients, the randomization process (if any), and how patients were assigned to each group.

Results:

The table can be improved by:

• Adding a description of the parameters in the header to provide context to the reader.

• Using a consistent formatting style for the values (e.g., use either square brackets or parentheses to denote ranges).

• Including units for all the parameters in the header, and making sure they are consistent throughout the table.

• Use asterisks or stars to denote significance levels.

Discussion:

• Line 56: Consider specifying the benefits of continuous infusion compared to intermittent administration in terms of pharmacokinetic/pharmacodynamic parameters.

• Provide a broader context for the results by discussing the implications of the study findings for the treatment of SCD and ACS.

• Emphasize the importance of the study results for improving the treatment of SCD and ACS.

7. PLOS authors have the option to publish the peer review history of their article (what does this mean?). If published, this will include your full peer review and any attached files.

Reviewer #4: No

Reviewer #5: **Yes: **Dr Victor Abiola Adepoju

---

## [Author Response · Author response to Decision Letter 3]

30 Mar 2023

We thank the reviewer for their constructive comments. All comments have been addressed and changes made in the manuscript. We have submitted a point-by-point response to all reviewers’ comments. 

All authors have read and approved the submission of the manuscript, and the manuscript has not been published and is not being considered for publication elsewhere in whole or in part in any language. Please, feel free to contact me for any requirements related to this submission.

Thank you so much for your attention.

Sincerely,

Keyvan Razazi, MD

Corresponding Author

Service de Réanimation Médicale, Hôpital Henri Mondor, Créteil, France

E-mail : keyvan.razazi@aphp.fr

Sincerely

Reviewer #5: This is a good paper that provided new insight and merit advancing to the next stage. However given the paucity of the data presented, I will suggest that it is best considered as a research letter rather than a full original article. 

Given the informations added as per reviewer’s comment and asked by the previous reviewers, we kept the manuscript as a research article. 

See my few improvement suggestions below:

Introduction:

• Line 64-73: The introduction could provide more background on the epidemiology of SCD and ACS and why these conditions are important to study.

Responses: In accordance with reviewer’s comment, a sentence was added as follows: Approximately, 300 000 infants are born annually worldwide with SCD and 100 000 people have SCD in the US. Acute chest syndrome (ACS) is a major complication of SCD and the first cause of mortality in adult patients.

• Add a brief overview of previous studies investigating the use of Cefotaxime in SCD and ACS. 

Responses: Cefotaxime was studied by Maksoud et al. only in children with SCD. However, data in adult patients are lacking. 

The reference of Maksoud et al is cited in the manuscript, and in accordance with reviwer’s comment, we added the following sentence: “Pharmacokinetic data on cefotaxime in adults patients with SCD are lacking.” 

• Discuss the current state of empirical antibiotic therapy in ACS and the need for further research in this area.

Responses: In accordance with reviewer comment, the discussion was amended as follows: ACS is characterized by fever and/or respiratory symptoms with new pulmonary infiltrates. Bacterial infection is documented in only a minority of cases [2,3]. However, empirical antibiotic therapy is often used in ACS because establishing a definitive etiology is not always possible. National guidelines and expert consensus strongly recommend antibiotic therapy for almost all patients with ACS despite the low quality of supporting evidence [1]. Further studies are needed to improve the selection of patients requiring antibiotic therapy and/or the early stop of empiric antibiotic therapy in this setting as untreated infection may be fatal within hours in patients with SCD.

Methods:

• Generally, the STROBE checklist for observational study should be followed for the manuscript including the method section

Responses: Methods were amended to fulfill SROBE checklist. STROBE checklist was added in OLS 

• Line 89-92: Consider specifying the inclusion and exclusion criteria for the study population, such as the criteria for being admitted to the ICU for severe ACS and age range of patients.

Responses: Inclusion and exclusion criteria were added as follows:

The inclusion criterion was a severe ACS episode requiring ICU with cefotaxime administration. Severe ACS was defined as previously reported by ICU admission (as decided on the basis of a collegiate clinical assessment by the referring physician at the SCD clinic and the intensivist) based on one of the following signs [7] : respiratory rate greater than 30 breaths/minute, increased respiratory accessory muscle activity, partial pressure of oxygen in arterial blood less than 60 mm Hg on room air, respiratory acidosis, altered consciousness, extensive parenchymal opacities on the chest radiograph, and multiple organ failures. Exclusion criteria were the use of antibiotic regimen without cefotaxime, antibiotic dosage not performed, age <18 years old, and patient’s refusal of data collected.

• Line 105-106: The methods for collecting data on hospital length of stay and infection relapse could be described in more detail.

Responses: In our center, patients are systematically seen in consultation after hospital discharge following an episode of ACS. All data were collected from patient’s chart and microbiology laboratory archives.

• Consider adding a power analysis to estimate the sample size necessary for the study. 

• Discuss the assumptions made when carrying out the statistical analysis.

Responses: The power analysis was already in the statistical section, as follows: “We estimated that a total of 30 consecutive patients per group was deemed necessary in order to observe an increase of the percentage of patients with a trough concentration of cefotaxime ≥2mg/L from 17% with IA [6], to 60% [8,9] with CI, considering a power of 90%, and an alpha risk of 5%.”

• Line 98-100: The methods for measuring Cefotaxime trough levels and residual concentrations could be described in more detail.

Responses: Blood samples were obtained just before infusion of the drug (trough concentrations) and after at least the fourth dose (IA) or at any time after 24 hours (CI). These points are mentioned in the methods section.

• Line 89-93: The design of the study could be described in more detail, including the inclusion and exclusion criteria for patients, the randomization process (if any), and how patients were assigned to each group.

Responses: Inclusion and exclusion were added (see above). As mentioned in the methods section, this study is a prospective before-after study, with no randomization. 

Results:

The table can be improved by:

• Adding a description of the parameters in the header to provide context to the reader.

Responses: ACS was defined in the title

• Using a consistent formatting style for the values (e.g., use either square brackets or parentheses to denote ranges).

Responses: As stated in the footnote in Table 1, data are presented as median [25th-75th percentiles] or number (percentage)

• Including units for all the parameters in the header, and making sure they are consistent throughout the table.

Responses: Units are now consistent throughout the table. 

• Use asterisks or stars to denote significance levels.

Responses: As proposed, stars are now used to denote significant p values.

Discussion:

• Line 56: Consider specifying the benefits of continuous infusion compared to intermittent administration in terms of pharmacokinetic/pharmacodynamic parameters.

Responses: This point was specified as follows: “The results of the DALI study support the conclusions that better outcomes for critically ill patients can be expected with higher drug exposures [9]. Prolonged infusion schemes increase the fraction of the dose interval in which unbound antibiotic concentrations exceed MIC of the pathogen (fT > MIC) compared with standard intermittent infusion[12–14]. A recent meta-analysis showed that hospital mortality was significantly lower and clinical cure was significantly higher in the continuous infusion group than in the intermittent infusion group[15]”

• Provide a broader context for the results by discussing the implications of the study findings for the treatment of SCD and ACS.

• Emphasize the importance of the study results for improving the treatment of SCD and ACS.

Responses: A sentence was added in the discussion section as follows: Our report suggests that other studies should be performed on the pharmacokinetic/pharmacodynamic of other treatments influenced by renal clearance in patients with SCD.

---

## [Decision Letter · Decision Letter 4]

10 Apr 2023

PONE-D-21-21735R4Decreased risk of underdosing with continuous infusion versus intermittent administration of Cefotaxime in patients with sickle cell disease and acute chest syndrome.PLOS ONE

Dear Dr. Razazi,

Thank you for submitting your manuscript to PLOS ONE. After careful consideration, we feel that it has merit but does not fully meet PLOS ONE’s publication criteria as it currently stands. Therefore, we invite you to submit a revised version of the manuscript that addresses the points raised during the review process.

Please revise. 

We look forward to receiving your revised manuscript.

Kind regards,

Academic Editor

PLOS ONE

Journal Requirements:

Reviewers' comments:

Reviewer's Responses to Questions

**Comments to the Author**

1. If the authors have adequately addressed your comments raised in a previous round of review and you feel that this manuscript is now acceptable for publication, you may indicate that here to bypass the “Comments to the Author” section, enter your conflict of interest statement in the “Confidential to Editor” section, and submit your "Accept" recommendation.

Reviewer #4: All comments have been addressed

Reviewer #5: (No Response)

2. Is the manuscript technically sound, and do the data support the conclusions?

Reviewer #4: Yes

Reviewer #5: Partly

3. Has the statistical analysis been performed appropriately and rigorously? 

Reviewer #4: Yes

Reviewer #5: No

4. Have the authors made all data underlying the findings in their manuscript fully available?

Reviewer #4: Yes

Reviewer #5: Yes

5. Is the manuscript presented in an intelligible fashion and written in standard English?

Reviewer #4: Yes

Reviewer #5: Yes

6. Review Comments to the Author

Reviewer #4: The authors of the manuscript "Decreased risk of underdosing with continuous infusion versus intermittent administration of Cefotaxime in patients with sickle cell disease and acute chest syndrome“ addressed all comments of the previous review.

Reviewer #5: Here are some feedback and improvement suggestions for the article titled "Decreased risk of underdosing with continuous infusion versus intermittent administration of cefotaxime in patients with sickle cell disease and acute chest syndrome" with line numbers:

Abstract:

The study population could be described more specifically with age range and number of participants.

Introduction:

The limitations of intermittent cefotaxime dosing and the potential consequences of underdosing could be better contextualized.

The rationale for the study could be more clearly stated.

Methods:

The recruitment process for patients could be described in more detail.

It should be clarified whether patients were randomized or not.

More information on the dosage regimen for each group could be provided.

It should be clarified whether the study was double-blinded or not.

The statistical methods used for data analysis could be described in more detail.

Results:

The difference in incidence of underdosing between the two groups should be clarified.

The numerical values for the percentages of underdosing in each group should be provided.

More detail on the adverse events observed in the study could be given.

Discussion:

The limitations of the study, including the small sample size and potential biases, should be discussed in more detail.

The clinical implications of the study results and potential for improving patient outcomes should be discussed in more detail.

Conclusion:

The conclusion could be more clearly stated with a summary of the study results and their clinical significance.

References: Some references are missing information such as authors' names or full journal titles, and some are not in proper citation format. These should be corrected.

Improvement suggestions: The introduction could be strengthened with more background information on ACS in SCD, its prevalence, and clinical management. The discussion could benefit from a comparison of the current study's results with other studies in the literature on cefotaxime dosing in SCD patients with ACS. A limitation of the study is that it only included patients with SCD and ACS, so the generalizability of the results to other patient populations may be limited. This limitation could be acknowledged and discussed in the discussion section. The article could benefit from a clearer explanation of the clinical implications of the study results for patient care and potential areas for future research. Consider reorganizing the methods section to improve clarity and readability. Consider adding a section on future directions and potential avenues for further research.

7. PLOS authors have the option to publish the peer review history of their article (what does this mean?). If published, this will include your full peer review and any attached files.

Reviewer #4: No

Reviewer #5: **Yes: **Adepoju Victor Abiola

---

## [Author Response · Author response to Decision Letter 4]

29 Apr 2023

We thank the reviewer for their constructive comments. All comments have been addressed and changes made in the manuscript. We have submitted a point-by-point response to all reviewers’ comments. 

All authors have read and approved the submission of the manuscript, and the manuscript has not been published and is not being considered for publication elsewhere in whole or in part in any language. Please, feel free to contact me for any requirements related to this submission.

Thank you so much for your attention.

Sincerely,

Keyvan Razazi, MD

Corresponding Author

Service de Réanimation Médicale, Hôpital Henri Mondor, Créteil, France

E-mail : keyvan.razazi@aphp.fr

Sincerely

Reviewer #5: Here are some feedback and improvement suggestions for the article titled "Decreased risk of underdosing with continuous infusion versus intermittent administration of cefotaxime in patients with sickle cell disease and acute chest syndrome" with line numbers:

Abstract:

The study population could be described more specifically with age range and number of participants.

Responses : In accordance with reviewer’s comment, a sentence was added as follows: We included 60 episodes of acute chest syndrome in 58 patients (29 [25-34] years, 37/58 (64%) males).

Introduction:

The limitations of intermittent cefotaxime dosing and the potential consequences of underdosing could be better contextualized. The rationale for the study could be more clearly stated.

Responses: In accordance with reviewer’s comment, the rationale was further detailed as follows: ß-lactams are the most commonly prescribed antimicrobial agents in the critically-ill, especially in patients with acute lung injury [4] . ß-lactam antibiotics display time-dependent activity, where bacterial killing (microbiologic cure) and treatment efficacy (clinical cure) correlate with the duration of time that free (unbound) plasma drug concentrations remain above the minimal inhibitory concentration of the offending pathogen (fT>MIC). These characteristics suggest optimal effects with continuous infusion (CI) rather than traditional intermittent administration (IA) dosing. IA dosing may result in ß-lactam concentrations below the MIC for much of the dosing interval, increasing the risk of treatment failure and emergence of resistant strains. In patients with sepsis, interventions aimed at optimising antibiotic therapy demonstrated the greatest improvements in clinical outcomes [4,5].

Methods:

The recruitment process for patients could be described in more detail.

It should be clarified whether patients were randomized or not.

More information on the dosage regimen for each group could be provided.

It should be clarified whether the study was double-blinded or not.

Responses: This before-after study was not randomized and was not double blinded. In accordance with reviewer’s comment, these points were further detailed in the methods as follows: In this before-after study, consecutive patients were included without randomization.

The dosage regimen was also further detailed, as follows:

The target dose of cefotaxime was 60 mg/kg/day for 7 days (or less according to procalcitonin concentrations) in both groups. IA used 30-60 min intravenous perfusions q6-8 hours [1g q8 hours (n=10); 2g q8 hours (n=7); 1g q6 hours (n=5); 1g q4 hours (n=8)]; , while CI used a 12-h infusion with an electric syringe pump, after a loading dose of 1g over 60 minutes.

The statistical methods used for data analysis could be described in more detail.

Responses: In accordance with reviewer’s comment, a sentence was added as follows:

Correlations were tested using the Spearman’s method.

Results:

The difference in incidence of underdosing between the two groups should be clarified.

The numerical values for the percentages of underdosing in each group should be provided.

Responses: In accordance with reviewer’s comment, numerical values and percentages were added in results section as follows:

Episodes with undetectable (<2 mg/L) trough concentrations were more frequent in IA group (n=25, 83%) than in CI group (n=2; 7%), p<0.01.

More detail on the adverse events observed in the study could be given.

Responses: In accordance with reviewer’s comment, adverse events were detailed as follows: We observed no adverse event associated with cefotaxime use, either neurologic (seizure, encephalopathy, coma, or delirium), hematological, or renal.

Discussion:

The limitations of the study, including the small sample size and potential biases, should be discussed in more detail.

Responses: In accordance with reviewer’s comment, limitations on sample size were added:

Our study has several limitations, including the small sample size, the assessment of a single antibiotic at a given dose, and the lack of clinical outcome. In addition, the before-after design and monocenter setting in an intensive care unit may infer several bias. Our results cannot be extrapolated to SCD patients with decreased renal function, to other antibiotics with a concentration-dependent therapeutic effect such as aminoglycosides or to patients without SCD. 

The clinical implications of the study results and potential for improving patient outcomes should be discussed in more detail.

Responses: In accordance with reviewer’s comment, we added this point as follows: Our findings have major clinical implications, given the burden of severe infections and difficult to treat infections in patients with SCD [21,22].

Conclusion:

The conclusion could be more clearly stated with a summary of the study results and their clinical significance.

Responses: In accordance with reviewer’s comment, the conclusion was changed as follows:

As compared to IA, CI of cefotaxime maximizes the pharmacokinetic/pharmacodynamic parameters. The median residual concentration was higher in the CI than IA group. CI of time-dependent antibiotics seems to decrease the risk of underdosing in patients with SCD. Daily nurse timework and cost were also lower with CI. Further studies are needed to assess the impact of CI on clinical outcomes in patients with SCD.

References: Some references are missing information such as authors' names or full journal titles, and some are not in proper citation format. These should be corrected.

Responses: We used the plos one style in Zotero 

Improvement suggestions: The introduction could be strengthened with more background information on ACS in SCD, its prevalence, and clinical management. 

Response: In accordance with reviewer suggestion, introduction was strengthened with data on prevalence of SCD and ACS, as well as clinical management, as follows: 

Sickle-cell disease (SCD) is one of the most common severe monogenic disorders worldwide. Approximately, 300,000 infants are born annually worldwide with SCD and 100,000 people have SCD in the US [1].

Acute chest syndrome (ACS) is a major complication of SCD and the first cause of mortality in adult patients [2]. ACS is characterized by fever and/or respiratory symptoms with new pulmonary infiltrates. Bacterial infection is documented in only a minority of cases [2,3]. However, empirical antibiotic therapy is often used in ACS because establishing a definitive etiology is not always possible.

The discussion could benefit from a comparison of the current study's results with other studies in the literature on cefotaxime dosing in SCD patients with ACS. 

Response: In accordance with reviewer’s comment, our study was compared to the only study from the literature on cefotaxime dosing in patients with SCD as follows: 

Our results are in accordance with those of Maksoud et al who found very low attainment of target cefotaxime serum concentrations in pediatric SCD patients (only 5.6% of patients achieved 90% of time above the MIC of 2mg/L) [23].

A limitation of the study is that it only included patients with SCD and ACS, so the generalizability of the results to other patient populations may be limited. This limitation could be acknowledged and discussed in the discussion section. 

Response: In accordance with reviewer’s comment, we added theses points as follows: Most patients presented with augmented renal clearance, a frequent feature in patients with SCD at baseline and during ACS. Our results cannot be extrapolated to SCD patients with decreased renal function, to other antibiotics with a concentration-dependent therapeutic effect such as aminoglycosides or to patients without SCD.

The article could benefit from a clearer explanation of the clinical implications of the study results for patient care and potential areas for future research. 

Response: In accordance with reviewer’s comment, we added this point as follows:

Our findings have major clinical implications, given the burden of severe infections and difficult to treat infections in patients with SCD [21,22]. 

Consider reorganizing the methods section to improve clarity and readability. 

Response: In accordance with reviewer’s comment, we reorganized the methods section with sub-sections for more clarity as follows: 

Consider adding a section on future directions and potential avenues for further research.

Response: In accordance with reviewer’s comment, we added these points as follows in the discussion section: Further studies are needed to assess the feasibility of this approach in medical wards. Further studies are also warranted to assess the effect of the optimization of pharmacokinetic/pharmacodynamic parameters on clinical outcomes in this setting, inasmuch as the rate of bacterial documentation is low during ACS. Our findings suggests exploring the pharmacokinetic/pharmacodynamic of other antibiotics (especially for bloodstream and difficult to treat infections like bone-joint infections [20]) and that of other treatments influenced by renal clearance (e.g., morphine) in patients with SCD.

---

## [Decision Letter · Decision Letter 5]

24 Jul 2023

PONE-D-21-21735R5Decreased risk of underdosing with continuous infusion versus intermittent administration of Cefotaxime in patients with sickle cell disease and acute chest syndrome.PLOS ONE

Dear Dr. Razazi,

Thank you for submitting your manuscript to PLOS ONE. After careful consideration, we feel that it has merit but does not fully meet PLOS ONE’s publication criteria as it currently stands. Therefore, we invite you to submit a revised version of the manuscript that addresses the points raised during the review process.

We look forward to receiving your revised manuscript.

Kind regards,

Academic Editor

PLOS ONE

Journal Requirements:

**Additional Editor Comments:**

Please revise.

Reviewers' comments:

Reviewer's Responses to Questions

**Comments to the Author**

1. If the authors have adequately addressed your comments raised in a previous round of review and you feel that this manuscript is now acceptable for publication, you may indicate that here to bypass the “Comments to the Author” section, enter your conflict of interest statement in the “Confidential to Editor” section, and submit your "Accept" recommendation.

Reviewer #6: All comments have been addressed

2. Is the manuscript technically sound, and do the data support the conclusions?

Reviewer #6: Partly

3. Has the statistical analysis been performed appropriately and rigorously? 

Reviewer #6: Yes

4. Have the authors made all data underlying the findings in their manuscript fully available?

Reviewer #6: No

5. Is the manuscript presented in an intelligible fashion and written in standard English?

Reviewer #6: Yes

6. Review Comments to the Author

Reviewer #6: This is a review of a reviewed article.Many comments have been made by previous reviewers. The authors have made an effort to provide responses to comments. Nevertheless, we have presented above the remarks to be corrected or to be improved. 1- AFFILIATION OF CERTAIN AUTHORS. The affiliations of the following authors have been entered in the wrong place: Anne Hulin, Armand Mekontso Dessap. 2- The authors administered a loading dose of cefotaxime in patients treated by continuous injection (CI). No justification was provided for this choice and also the influence that this loading dose may have on the concentration of the drug in this group. 3- Evidence of infection was found in one patient. But there is no microbiological information on the identified germ. 4- The authors did not take into account the influence of the glomerular filtration rate on the concentration of the drug. A high filtration rate is usually accompanied by an increase in clearance of the drug, which may decrease its concentration. The authors could take this variable in consideration in the analysis of drug concentrations in the two groups and the resulting interpretation.

7. PLOS authors have the option to publish the peer review history of their article (what does this mean?). If published, this will include your full peer review and any attached files.

Reviewer #6: No

---

## [Author Response · Author response to Decision Letter 5]

27 Jul 2023

We thank the reviewer for their constructive comments. All comments have been addressed and changes made in the manuscript. We have submitted a point-by-point response to all reviewers’ comments. We hope that our manuscript is now suitable for publication in the journal.

All authors have read and approved the submission of the manuscript, and the manuscript has not been published and is not being considered for publication elsewhere in whole or in part in any language. Please, feel free to contact me for any requirements related to this submission.

Thank you so much for your attention.

Sincerely,

Keyvan Razazi, MD

Corresponding Author

Service de Réanimation Médicale, Hôpital Henri Mondor, Créteil, France

E-mail : keyvan.razazi@aphp.fr

Sincerely

 

Reviewer #6: This is a review of a reviewed article..Many comments have been made by previous reviewers. The authors have made an effort to provide responses to comments. Nevertheless, we have presented above the remarks to be corrected or to be improved. 

1- AFFILIATION OF CERTAIN AUTHORS. The affiliations of the following authors have been entered in the wrong place: Anne Hulin, Armand Mekontso Dessap. 

Response: We apologize for the typo error. In accordance with reviewer’s comment, affiliation have been corrected. 

2- The authors administered a loading dose of cefotaxime in patients treated by continuous injection (CI). No justification was provided for this choice and also the influence that this loading dose may have on the concentration of the drug in this group. 

Response: We used the CI protocols as proposed in septic patients to improve the pharmacokinetic/pharmacodynamic targets. The surviving sepsis campaign recommended : i) “For adults with sepsis or septic shock, we suggest using prolonged infusion of beta-lactams for maintenance (after an initial bolus) over conventional bolus infusion”; ii) “Administration of a loading dose of antibiotic before prolonged infusion is essential to avoid delays to achieving effective beta-lactam concentrations.” 

Of note, antibiotic dosage was performed a median of 3 days after cefotaxime initiation. Therefore, the loading dose performed to achieve effective beta-lactam concentrations could not explain on its own the difference of concentration between groups.

In accordance with reviewer’s comment, we performed the following changes: 

- We added the following sentence in the methods section: “A loading dose of antibiotic before prolonged infusion is recommended to avoid delays in achieving effective beta-lactam concentrations in critically-ill patients.”

- We added the time from cefotaxime initiation to antibiotic dosage in table 3 in the results section.

- We added the following sentence in the discussion section: “The loading dose performed to achieve effective beta-lactam concentrations could not explain on its own the difference of concentration between groups as dosage were performed 3 days after cefotaxime initiation.”

3- Evidence of infection was found in one patient. But there is no microbiological information on the identified germ. 

Response: In accordance with reviewer’s comment, microbioloical information was added as follows: “third-generation cephalosporin-susceptible Escherichia coli”

4- The authors did not take into account the influence of the glomerular filtration rate on the concentration of the drug. A high filtration rate is usually accompanied by an increase in clearance of the drug, which may decrease its concentration. The authors could take this variable in consideration in the analysis of drug concentrations in the two groups and the resulting interpretation.

Response: We thank the reviewer for this comment. Augmented renal clearance (ARC) is frequent among patients with sickle cell disease. The estimated glomerular filtration rate (using the CKD-EPI equation without adjustment for ethnicity [1]) and the frequency of augmented renal clearance were not different between groups (table 1). 

In accordance with reviewer’s comment, we also assessed the effect of CI while adjusting on estimated glomerular filtration rate and found consistent results. 

Sentences were added in methods and results section accordingly as follows: 

- Methods section: “The effect of CI on cefotaxime trough concentration ≥2 mg/L was also analyzed after adjustment on estimated glomerular filtration rate.”

- Result section: “This result persisted after adjustment on estimated glomerular filtration: crude and adjusted odds ratio (95% confidence interval) of 70 (12-393), p<0.001 and 111 (15-818), p<0.001, respectively).

1. Arlet J-B, Ribeil J-A, Chatellier G, Eladari D, De Seigneux S, Souberbielle J-C, et al. Determination of the best method to estimate glomerular filtration rate from serum creatinine in adult patients with sickle cell disease: a prospective observational cohort study. BMC Nephrology. 2012;13. doi:10.1186/1471-2369-13-83

---

## [Decision Letter · Decision Letter 6]

2 Oct 2023

PONE-D-21-21735R6Decreased risk of underdosing with continuous infusion versus intermittent administration of Cefotaxime in patients with sickle cell disease and acute chest syndrome.PLOS ONE

Dear Dr. Razazi,

Thank you for submitting your manuscript to PLOS ONE. After careful consideration, we feel that it has merit but does not fully meet PLOS ONE’s publication criteria as it currently stands. Therefore, we invite you to submit a revised version of the manuscript that addresses the points raised during the review process.

Please revise.

We look forward to receiving your revised manuscript.

Kind regards,

Academic Editor

PLOS ONE

Reviewers' comments:

Reviewer's Responses to Questions

**Comments to the Author**

1. If the authors have adequately addressed your comments raised in a previous round of review and you feel that this manuscript is now acceptable for publication, you may indicate that here to bypass the “Comments to the Author” section, enter your conflict of interest statement in the “Confidential to Editor” section, and submit your "Accept" recommendation.

Reviewer #7: All comments have been addressed

Reviewer #8: (No Response)

2. Is the manuscript technically sound, and do the data support the conclusions?

Reviewer #7: Partly

Reviewer #8: Yes

3. Has the statistical analysis been performed appropriately and rigorously? 

Reviewer #7: Yes

Reviewer #8: Yes

4. Have the authors made all data underlying the findings in their manuscript fully available?

Reviewer #7: Yes

Reviewer #8: Yes

5. Is the manuscript presented in an intelligible fashion and written in standard English?

Reviewer #7: Yes

Reviewer #8: Yes

6. Review Comments to the Author

Reviewer #7: 1- The aim of this study is not strong and also is not logical.

2- What was benefit of continues drug injection? more care during drug infusion and long stay at hospital. more expensive for patients.

3- How much does patient received Cefotaxime in continues or intermittent method?

4- It has been necessary to show kidney function at least 14 days after continues injection.

5- LFT (liver function test) in both method is necessary and should be compare and discus.

Reviewer #8: Razazi et al. present a paper in which they investigate and compare continuous injection (CI) vs intermittent administration (AI) of the antibiotic cefotaxime in sickle cell disease (SCD) patients presenting with acute chest syndrome. They find CI superior to AI in terms of pharmacokinetics and pharmacodynamics of the drug. In my opinion, and regarding their responses to the previous reviewers, this work provides some helpful insight into the clinical application of this antibiotic for SCD patients. However, I recommend the journal consider this as letter to the editor, brief communication, or things alike, not a full paper.

7. PLOS authors have the option to publish the peer review history of their article (what does this mean?). If published, this will include your full peer review and any attached files.

Reviewer #7: No

Reviewer #8: No

---

## [Author Response · Author response to Decision Letter 6]

7 Nov 2023

We thank the reviewer for their constructive comments. All comments have been addressed and changes made in the manuscript. We have submitted a point-by-point response to all reviewers’ comments. 

We would like to bring to the editor's attention that it has now been two and a half years since this paper has been submitted to your journal, with a total of 6 reviewing sessions involving 8 reviewers, and only minor corrections requested in the last round. Therefore, we deeply hope that at this very advanced stage of the reviewing process, after 2 and a half years, an ultimate and favorable decision will be made.

All authors have read and approved the submission of the manuscript, and the manuscript has not been published and is not being considered for publication elsewhere in whole or in part in any language. Please, feel free to contact me for any requirements related to this submission.

Thank you so much for your attention.

Sincerely,

Keyvan Razazi, MD

Corresponding Author

Service de Réanimation Médicale, Hôpital Henri Mondor, Créteil, France

E-mail : keyvan.razazi@aphp.fr

Sincerely

 

Reviewer #7: 

1- The aim of this study is not strong and also is not logical.

Response: 

This study was logical. Underdosing of antibiotics is common in patients with sickle cell disease (SCD), who often present augmented renal clearance secondary to increased renal perfusion. The use of continuous infusions (CI) could improve the pharmacokinetic/ pharmacodynamic targets as compared to intermittent administration (IA). So assessment of CI versus IA during acute chest syndrome, the most severe complication of SCD is logical. 

Indeed, the evaluation criterion for our study is not as strong as mortality, but the latter is relatively low during episodes of acute chest syndrome in patients with SCD. Because antibiotic are frequently prescribed in SCD, antibiotic stewardship (including the adequate dose and administration) are important in this setting.

2- What was benefit of continues drug injection? more care during drug infusion and long stay at hospital. more expensive for patients.

Response: As showed in the results section, daily nurse timework and cost were lower with CI. Median length of stay was similar between groups.

3- How much does patient received Cefotaxime in continues or intermittent method?

Response: As described in methods and results section, 30 patients received CI and 30 received IA. 

4- It has been necessary to show kidney function at least 14 days after continues injection.

Response: Renal function did not differ between groups at day 7 or day 14 after antibiotic initiation. In accordance with reviewer’s comment, these data were added in table 1 and results section.

5- LFT (liver function test) in both method is necessary and should be compare and discus.

Response: Only a small amount (2%) of cefotaxime is excreted through the liver. Liver function tests were not different between groups. Data were added in table 1 and in results section. 

Reviewer #8: Razazi et al. present a paper in which they investigate and compare continuous injection (CI) vs intermittent administration (AI) of the antibiotic cefotaxime in sickle cell disease (SCD) patients presenting with acute chest syndrome. They find CI superior to AI in terms of pharmacokinetics and pharmacodynamics of the drug. In my opinion, and regarding their responses to the previous reviewers, this work provides some helpful insight into the clinical application of this antibiotic for SCD patients. 

However, I recommend the journal consider this as letter to the editor, brief communication, or things alike, not a full paper.

There is no option for “brief communication or things alike” in the submission process. We would like to bring to the editor's and reviewer’s attention that it has now been two and a half years since this paper has been submitted to your journal, with a total of 6 reviewing sessions involving 8 reviewers, and only minor corrections requested in the last round. Therefore, we deeply hope that at this very advanced stage of the reviewing process, after 2 and a half years, an ultimate and favorable decision will be made.

---

## [Decision Letter · Decision Letter 7]

12 Feb 2024

PONE-D-21-21735R7Decreased risk of underdosing with continuous infusion versus intermittent administration of Cefotaxime in patients with sickle cell disease and acute chest syndrome.PLOS ONE

Dear Dr. Razazi,

Thank you for submitting your manuscript to PLOS ONE. After careful consideration, we feel that it has merit but does not fully meet PLOS ONE’s publication criteria as it currently stands. Therefore, we invite you to submit a revised version of the manuscript that addresses the points raised during the review process.

Please revise.

We look forward to receiving your revised manuscript.

Kind regards,

Academic Editor

PLOS ONE

Journal Requirements:

Reviewers' comments:

Reviewer's Responses to Questions

**Comments to the Author**

1. If the authors have adequately addressed your comments raised in a previous round of review and you feel that this manuscript is now acceptable for publication, you may indicate that here to bypass the “Comments to the Author” section, enter your conflict of interest statement in the “Confidential to Editor” section, and submit your "Accept" recommendation.

Reviewer #9: All comments have been addressed

Reviewer #10: (No Response)

2. Is the manuscript technically sound, and do the data support the conclusions?

Reviewer #9: Yes

Reviewer #10: Yes

3. Has the statistical analysis been performed appropriately and rigorously? 

Reviewer #9: Yes

Reviewer #10: Yes

4. Have the authors made all data underlying the findings in their manuscript fully available?

Reviewer #9: Yes

Reviewer #10: Yes

5. Is the manuscript presented in an intelligible fashion and written in standard English?

Reviewer #9: Yes

Reviewer #10: Yes

6. Review Comments to the Author

Reviewer #9: (No Response)

Reviewer #10: Peer Review Report for PONE-D-21-21735R7

Overall: The authors chose to conduct a study which evaluated the effect of continuous infusion over intermittent administration of cefotaxime in patients with sickle cell disease who have ACS. The authors specifically evaluated the trough level of cefotaxime, which was significantly better after CI. Although the authors did not see a clinical benefit, they reported a time-saving benefit, and a cost-benefit, whose relevance to practical use is questionable owing to the benefit being extremely small.

I understand that this paper has been reviewed by several reviewers before me and has been revised seven times. Therefore, and to the authors' credit, the quality of the manuscript was acceptable, and I only have small comments to offer. Although the authors chose to evaluate a very narrow, and specific, aspect of drug dosing, nevertheless, they have produced research which is not erroneous. And although there are no clinical benefits of this study, the authors have indeed answered a clinical question which hitherto had not been addressed before.

***Abstract***

[1] Page 3. Under the "Patients" subheading, the authors currently report the number of severe ACS "episodes" they encountered. Although this is relevant given the study was centered around these episodes, the author should also quote the number of patients as they do under the next subheading.

[2] Page 4. The authors conclude that CI helped cefotaxime achieve a better pharmacokinetic/pharmacodynamic profile. While this is true, and reflects the original purpose of the study, it is advisable that the authors also highlight that this did not translate into better clinical outcomes when compared to IA (similar hospital stays etc.). This would help future readers understand the wider picture without having to dig for data within the manuscript.

***Introduction***

[1] I have no particular comments for this section. The introduction flows well and is well-written. Although this is not a strict comment, but I would have liked to see the authors highlight other drugs which are also employed in ACS. A sentence about their use, and then the rationale for choosing cefotaxime would have been interesting to see. As of now, the introduction does justice to the research question at hand.

***Methods***

[1] Page 9. Time and Cost. Currently, the authors simply state that they calculated the cost of instruments, and labor etc.; and the time needed by observing ten nurses. However, this does not convey how this was achieved. As a researcher who would like to replicate this study, it offers no information on how this data was collected. As of now, we just know that it was collected. The authors should consider adding sentences that state all this data was collected. For example, what instruments were considered when calculating the cost (infusion pump? medicines? et al.), did the authors review institutional records to identify the cost? If so, it should be stated. When calculating the time spent, who recorded the time that was taken? And how?

I hope the authors understand that one of the pillars of rigorous research is reproducibility, which requires that all information regarding the research should be available. Therefore, to make this paper more scientifically accurate, while it may sound inconsequential, and to maintain scientific rigor it is necessary that such details be highlighted.

***Discussion***

[1] Page 15. The authors say "Moreover, nurse timework and costs were decreased with CI as previously published with other antibiotics". The authors should consider discussing whether saving 44 seconds by using CI provided an actual clinical/logistical benefit or not, as well as its importance.

***Conclusion***

[1] Page 16. Conclusion. The authors do not state that there was no clinical benefit of CI over IA. The authors should consider doing so to convey the clinical relevance of this study.

7. PLOS authors have the option to publish the peer review history of their article (what does this mean?). If published, this will include your full peer review and any attached files.

Reviewer #9: No

Reviewer #10: **Yes: **Muhammad Abdul Rehman

---

## [Author Response · Author response to Decision Letter 7]

14 Mar 2024

We thank the reviewer for their constructive comments. All comments have been addressed and changes made in the manuscript. We have submitted a point-by-point response to all reviewers’ comments. 

We would like to bring to the editor's attention that it has now been two and a half years since this paper has been submitted to your journal, with a total of eight reviewing sessions involving at least ten reviewers, and only minor corrections requested in the last round. Therefore, we deeply hope that at this very advanced stage of the reviewing process, after 30 months, an ultimate and favorable decision will be made.

All authors have read and approved the submission of the manuscript, and the manuscript has not been published and is not being considered for publication elsewhere in whole or in part in any language. Please, feel free to contact me for any requirements related to this submission.

Thank you so much for your attention.

Sincerely,

Keyvan Razazi, MD

Corresponding Author

Service de Réanimation Médicale, Hôpital Henri Mondor, Créteil, France

E-mail : keyvan.razazi@aphp.fr

Sincerely

Overall: The authors chose to conduct a study which evaluated the effect of continuous infusion over intermittent administration of cefotaxime in patients with sickle cell disease who have ACS. The authors specifically evaluated the trough level of cefotaxime, which was significantly better after CI. Although the authors did not see a clinical benefit, they reported a time-saving benefit, and a cost-benefit, whose relevance to practical use is questionable owing to the benefit being extremely small.

I understand that this paper has been reviewed by several reviewers before me and has been revised seven times. Therefore, and to the authors' credit, the quality of the manuscript was acceptable, and I only have small comments to offer. Although the authors chose to evaluate a very narrow, and specific, aspect of drug dosing, nevertheless, they have produced research which is not erroneous. And although there are no clinical benefits of this study, the authors have indeed answered a clinical question which hitherto had not been addressed before.

***Abstract***

[1] Page 3. Under the "Patients" subheading, the authors currently report the number of severe ACS "episodes" they encountered. Although this is relevant given the study was centered around these episodes, the author should also quote the number of patients as they do under the next subheading.

Responses: In accordance with reviewer’s comment, the number of patients was added in patients suheading as follows: “Sixty consecutive episodes of severe acute chest syndrome in 58 adult patients with sickle cell disease.”

[2] Page 4. The authors conclude that CI helped cefotaxime achieve a better pharmacokinetic/pharmacodynamic profile. While this is true, and reflects the original purpose of the study, it is advisable that the authors also highlight that this did not translate into better clinical outcomes when compared to IA (similar hospital stays etc.). This would help future readers understand the wider picture without having to dig for data within the manuscript.

Responses: In accordance with reviewer’s comment, a sentence was added in the conclusion of the abstract as follows: “The clinical outcome did not differ between the two administration methods; however, the study was underpowered to detect such a difference.”

***Introduction***

[1] I have no particular comments for this section. The introduction flows well and is well-written. Although this is not a strict comment, but I would have liked to see the authors highlight other drugs which are also employed in ACS. A sentence about their use, and then the rationale for choosing cefotaxime would have been interesting to see. As of now, the introduction does justice to the research question at hand.

Responses: In accordance with reviewer’s comment , a sentence was added as follows: “In clinical practice, empirical intravenous antimicrobial therapy is usually a combination therapy with a macrolide targeting intracellular bacteria and a beta lactam targeting pyogenic bacteria (amoxicillin in mild ACS and cephalosporin for severe ACS) [4].” 

***Methods***

[1] Page 9. Time and Cost. Currently, the authors simply state that they calculated the cost of instruments, and labor etc.; and the time needed by observing ten nurses. However, this does not convey how this was achieved. As a researcher who would like to replicate this study, it offers no information on how this data was collected. As of now, we just know that it was collected. The authors should consider adding sentences that state all this data was collected. For example, what instruments were considered when calculating the cost (infusion pump? medicines? et al.), did the authors review institutional records to identify the cost? If so, it should be stated. When calculating the time spent, who recorded the time that was taken? And how?

I hope the authors understand that one of the pillars of rigorous research is reproducibility, which requires that all information regarding the research should be available. Therefore, to make this paper more scientifically accurate, while it may sound inconsequential, and to maintain scientific rigor it is necessary that such details be highlighted.

Responses: As requested by the reviewer, the methods for time and costs were detailed in the methods section and in the online supplement as follows: 

“The time and associated costs (drug and consumables) of administering a continuous versus an intermittent infusion were compared. One author (KR) observed 10 nurses administering, in succession, in a simulation room, a continuous and an intermittent infusion in a cross-over design (five nurses began by the continuous infusion while five others began by the intermittent infusion). The time needed to prepare and administer the antibiotic was calculated for each infusion method (see online supplement). This time was collected using a stopwatch, and included the preparation phases in the patient's room, tubing connection, start of infusion, stop of infusion, and tubing disconnection. The time dedicated to collecting equipment was not taken into account because most of the material needed was located in the room. The time needed per day to prepare and administer the antibiotic was computed as the product of the time for one administration by the number of administrations per day.

The costs for all items used in drug preparation and administration including syringe pump amortization expenses were also calculated (see online supplement). The costs were identified by reviewing institutional records (tax included). These costs included: cefotaxime, crystalloid sodium 0,9% 50ml, transfert needle, sterile dressings, syringe with luer lock, extension set (only for continuous infusion) or infusion set (only for intermittent administration), and syringe pump amortization expenses (only for continuous infusion). Labor cost of the nursing time was not considered.”

Of note, when revising the costs, we corrected a calculating error, not affecting the trend of the results: Cost of all necessary material and drug were lower with CI as compared to IA: 6.1 [6.1-7.0] vs 7.8 [5.6-11.1] euros, p<0.01. Sorry for this mistake.

***Discussion***

[1] Page 15. The authors say "Moreover, nurse timework and costs were decreased with CI as previously published with other antibiotics". The authors should consider discussing whether saving 44 seconds by using CI provided an actual clinical/logistical benefit or not, as well as its importance.

Response: In accordance with reviewer’s comment, we added the following sentences: “The time saved for nurses was due to the reduction in the number of administrations per day. However, this result was obtained in a simulation room, and the difference was numerically marginal (less than one minute).”

***Conclusion***

[1] Page 16. Conclusion. The authors do not state that there was no clinical benefit of CI over IA. 

Response: In accordance with reviewer comment, a sentence was added in the conclusion of the manuscript as follows: “The clinical outcome did not differ between the two administration methods; however, the study was underpowered to detect such a difference.”

---

## [Decision Letter · Decision Letter 8]

2 Apr 2024

Decreased risk of underdosing with continuous infusion versus intermittent administration of Cefotaxime in patients with sickle cell disease and acute chest syndrome.

PONE-D-21-21735R8

Dear Dr. Razazi,

We’re pleased to inform you that your manuscript has been judged scientifically suitable for publication and will be formally accepted for publication once it meets all outstanding technical requirements.

Kind regards,

Academic Editor

PLOS ONE

Additional Editor Comments (optional):

Reviewers' comments:

Reviewer's Responses to Questions

**Comments to the Author**

1. If the authors have adequately addressed your comments raised in a previous round of review and you feel that this manuscript is now acceptable for publication, you may indicate that here to bypass the “Comments to the Author” section, enter your conflict of interest statement in the “Confidential to Editor” section, and submit your "Accept" recommendation.

Reviewer #9: All comments have been addressed

Reviewer #10: All comments have been addressed

2. Is the manuscript technically sound, and do the data support the conclusions?

Reviewer #9: Yes

Reviewer #10: Yes

3. Has the statistical analysis been performed appropriately and rigorously? 

Reviewer #9: Yes

Reviewer #10: I Don't Know

4. Have the authors made all data underlying the findings in their manuscript fully available?

Reviewer #9: Yes

Reviewer #10: Yes

5. Is the manuscript presented in an intelligible fashion and written in standard English?

Reviewer #9: Yes

Reviewer #10: Yes

6. Review Comments to the Author

Reviewer #9: The manuscript is probably revised and i recommend to publish it in your journal. The author probably designed the manuscript well

Reviewer #10: I have gone through the revised manuscript and am satisfied with the revisions the authors have made. I believe all my comments have been addressed.

7. PLOS authors have the option to publish the peer review history of their article (what does this mean?). If published, this will include your full peer review and any attached files.

Reviewer #9: No

Reviewer #10: **Yes: **Muhammad Abdul Rehman

---

## [Editor Report · Acceptance letter]

4 Apr 2024

PONE-D-21-21735R8 

PLOS ONE

Dear Dr. Razazi, 

I'm pleased to inform you that your manuscript has been deemed suitable for publication in PLOS ONE. Congratulations! Your manuscript is now being handed over to our production team.

Kind regards, 

on behalf of

Dr. Robert Jeenchen Chen 

Academic Editor

PLOS ONE